# All-round catalytic and atroposelective strategy via dynamic kinetic resolution for N-/2-/3-arylindoles

Ahreum Kim[1], Chanhee Lee[1], Jayoung Song[2], Sang Kook Lee [2] &
Yongseok Kwon [1] ✉

As the complexity of organic molecules utilized by mankind increases, the phenomenon of atropisomerism is more frequently encountered. While a variety of well-established methods enable the control of a stereogenic center, a catalytic method for controlling a stereogenic axis in one substrate is typically unavailable for controlling axial chirality in other substrates with a similar structure. Herein, we report *o*-amidobiaryl as a flexible platform for chiral phosphoric acid-catalyzed atroposelective dynamic kinetic resolution. To demonstrate our strategy, three distinct types of arylindoles were utilized and reacted intermolecularly with ketomalonate in the presence of chiral phosphoric acid. An investigation of 46 substrates having an aromatic ring in different positions yields the desired products with excellent enantioselectivities. Computational investigation into the origin of enantioselectivity highlights the importance of the NH group. Given the biological significance of indoles, antiproliferative effects have been investigated; our scaffold exhibits good efficacy in this regard.

Novel catalytic and asymmetric reactions are being developed continuously, which compensate for the drawbacks in previous methods and make new chiral molecules easily accessible[1–4]. In particular, for the control of a stereogenic center, several catalytic and reliable methods have been developed with the least required functional group in a substrate, owing to the accumulation of numerous valuable precedents[5–7]. However, controlling several types of chiralities, such as axial[8–10], helical[11,12], and planar chiralities[13,14], remains difficult; therefore, even a minor modification in a substrate frequently causes a substantial loss of yields or selectivities[15,16]. Thus, even though newly developed individual methods represent the cutting-edge of asymmetric catalysis, a versatile strategy that can control a difficult stereogenic element with a high degree of generality would be ideal and desirable[15,16].

In well-established asymmetric reactions, secondary interactions between asymmetric catalysts and substrates frequently play a significant role in achieving high enantioselectivity[6,7,17,18]. For example, in the Sharpless asymmetric epoxidation, a titanium-coordinated hydroxyl group close to an alkene confers facial selectivity (Fig. 1a, top)[19]. In addition, β-ketoesters are recognized as the optimal substrate for the Noyori asymmetric hydrogenation, because of the interaction between the metal and the carbonyl oxygen (Fig. 1a, bottom)[20]. These illustrative examples suggest that introducing a novel and flexible secondary interaction would create a new path, particularly in the challenging topics of asymmetric catalysis. In this regard, the present study proposes a general strategy for atroposelective dynamic kinetic resolution[21–34] in which one aromatic ring is substituted by the aid of a catalytic chiral phosphoric acid[35–37] while the other aromatic ring forms a secondary interaction with the catalyst[38–40] (Fig. 1b).

Highly substituted indoles play a crucial role in organic chemistry as essential structural motifs in natural products[41], biologically active compounds[42,43], and material sciences[44]. Due to the higher electron density of a nitrogen-containing five-membered ring (A ring in Fig. 1c) compared to a carbocyclic ring (B ring in Fig. 1c)[45], substitutions are

[1]School of Pharmacy, Sungkyunkwan University, Suwon 16419, Republic of Korea. [2]College of Pharmacy, Seoul National University, Seoul 08826, Republic of Korea. ✉e-mail: y.kwon@skku.edu

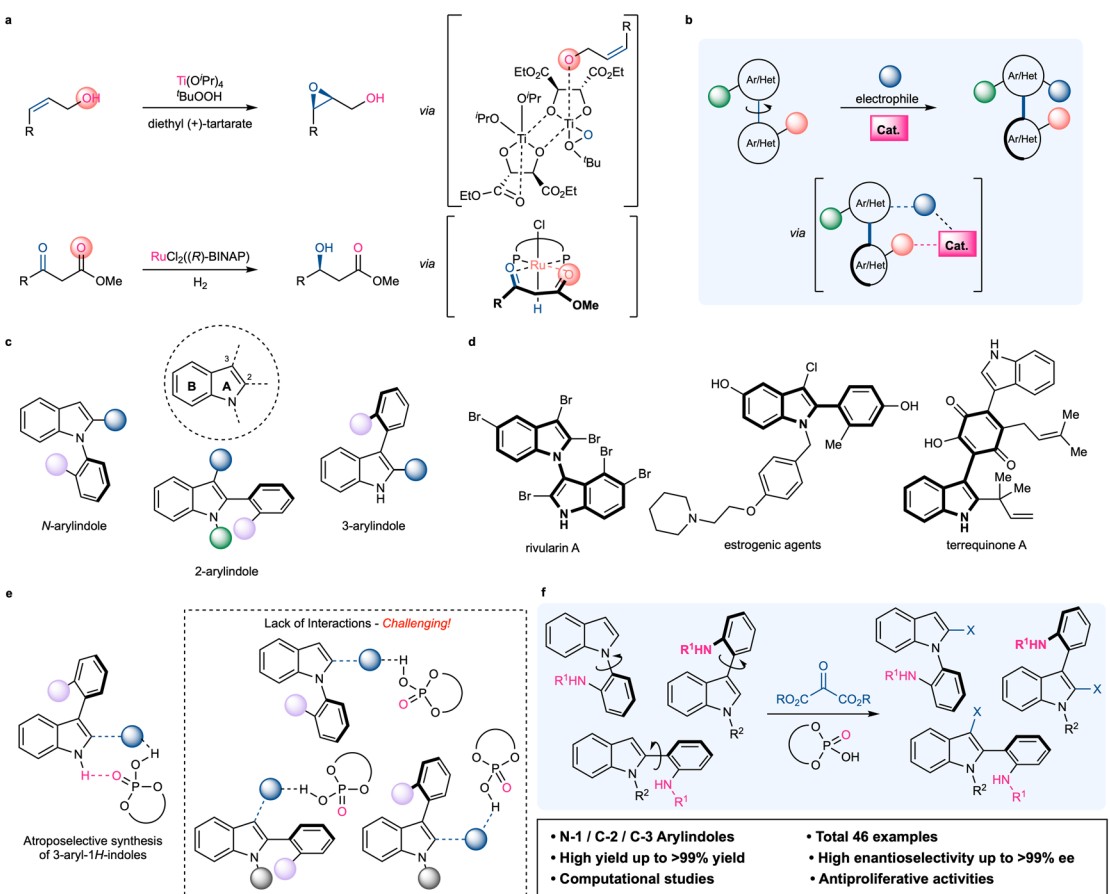

**Fig. 1 | Development of a general strategy for catalytic atroposelective synthesis. a** Examples of key secondary interactions in asymmetric catalysis. **b** General concept.
**c** Atropisomerism in arylindoles. **d** Examples of atropisomeric arylindoles. **e** Catalytic atroposelective synthesis of substituted arylindoles. **f** This work.

typically concentrated at the $N$−1, $C$−2, and $C$−3 positions. Specifically, the introduction of an aromatic ring at these positions has significantly elongated the indole scaffold both in nature and in the laboratory; consequently, the discovery of atropisomerism in indoles continues to increase (Fig. 1d)[46–48]. Due to its biological[49] and synthetic significance[50], numerous efforts have been directed to the catalytic control of a stereogenic axis in indoles, which represents the state of the art in asymmetric catalysis[51–57]. Among them, catalytic dynamic kinetic resolutions of 3-aryl-1$H$-indoles have been investigated because of the high reactivity at the $C$−2 position (Fig. 1e). For example, the Shi group reported an atroposelective intermolecular functionalization of 3-arylindoles with multiple electrophiles catalyzed by chiral phosphoric acid[54–56]. More recently, the Fu group employed indole-3-one as an electrophile to control the stereogenic center and axis[57]. They hypothesized that the interaction between the N−H in the indole and the P=O in chiral phosphoric acid is essential for achieving high enantioselectivity[38–40]. While these findings have provided general and efficient for synthesizing atropisomeric 3-arylindoles, they have intrinsic limitations for other arylindoles that lack interaction, such as $N$-arylindole, 2-arylindole, and $N$-substituted-3-arylindole (Fig. 1e).

Our group recently reported the chiral phosphoric acid−catalyzed atroposelective Pictet−Spengler cyclization of $N$-arylindoles and biphenyls[58,59]. In this reaction, an N−H group was introduced to a phenyl ring, creating a critical interaction with the catalyst. With our ongoing research interest in catalytic and atroposelective reactions, we hypothesized that intermolecular atroposelective functionalization would be realized by the introduction of an N−H group into a substrate. Furthermore, $o$-amino/$o$-amidobiaryl substrates could be extensively utilized as a platform for catalytic and atroposelective

dynamic kinetic resolution. To illustrate our theory, we envisioned the atroposelective synthesis of not only $N$-arylindoles, but also 2-arylindoles and 3-arylindoles. To the best of our knowledge, one catalytic method that can control an axis at three separate positions of indoles has not been developed. Moreover, atroposelective synthesis of 2-arylindole via dynamic kinetic resolution is rare. Herein, we provide a highly atroposelective dynamic kinetic resolution of three distinct types of arylindoles with ketomalonates catalyzed by a chiral phosphoric acid, thereby providing an effective method for indoles with a high degree of generality (Fig. 1f).

## Results
### Catalytic dynamic kinetic resolution of $N$-/2-/3-arylindoles
Generally, atropisomerism occurs when at least three substituents surround a stereogenic single bond[60,61]. To demonstrate our general concept shown in Fig. 1b, $N$-arylindole, 2-arylindole, and 3-arylindole containing a benzamide group at the $ortho$ position of the phenyl rings were designed. We hypothesized that the catalytic functionalization at the $C$−2 position of $N$-arylindole and 3-arylindole would create a sufficient rotational barrier to maintain stereo-configuration. However, because the remaining $N$−1 and $C$−3 positions are more reactive than the $C$−2 position, the methyl group was introduced to the $C$−3 position of $N$-arylindole (**1a**) and $N$−1 position of 3-arylindole (**6a**). After the catalytic reaction at the $C$−3 position, we anticipated that a methyl group would be required at the $N$−1 position of 2-arylindole (**4a**) to achieve a high rotational barrier. We believe that these three regioisomeric substrates (**1a**, **4a**, and **6a**) demonstrate the broad applicability of our technique.

To test our hypothesis, chiral phosphoric acids were screened (Fig. 2) in chloroform at 40 °C with $N$-arylindole (**1a**) and diethyl

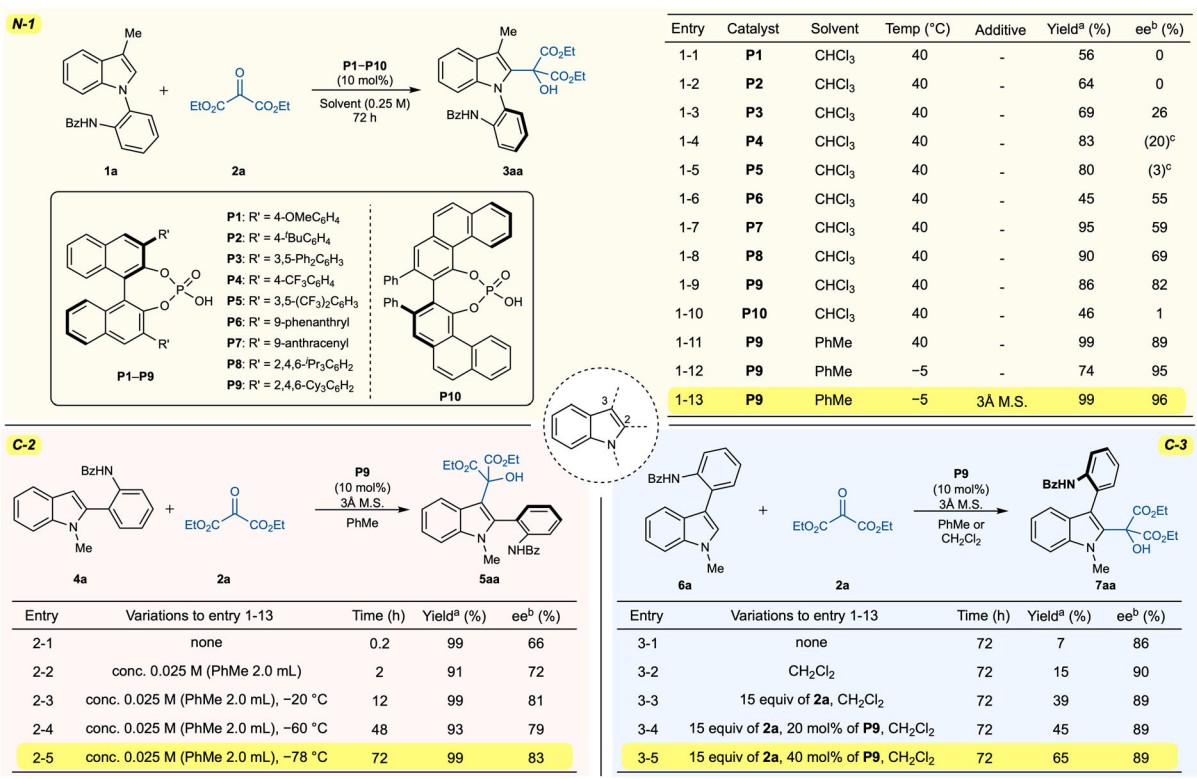

**Fig. 2 | Selected optimizations of the reaction conditions.** [a]Isolated yields. [b]Enantiomeric excesses were determined via chiral high-performance liquid chromatography analysis. [c]The values in parentheses represent the enantiomeric excesses of the opposite enantiomer.

ketomalonate (**2a**) (For the detailed electrophile screening, see the Supplementary Information). Initially, we examined chiral phosphoric acids (**P1**) substituted with 4-methoxyphenyl groups, which smoothly induced the reaction to produce the desired product (**3aa**) with a 56% yield despite the absence of enantioselectivity (Fig. 2, entry 1–1). The replacement of the 4-methoxy group with the 4-*tert*-butyl group did not enhance enantioselectivity (Fig. 2, entry 1–2). The reaction of **1a** with the 3,5-diphenyl-substituted catalyst (**P3**) exhibited marginally increased enantioselectivity (26% ee), indicating that the number and position of substituents play a crucial role in determining selectivity (Fig. 2, entry 1–3). Interestingly, the catalysts (**P4** and **P5**) bearing electron-withdrawing groups produced the opposite atropisomer to that obtained with **P3** (Fig. 2, entries 1–4 and 1–5), where the 4-trifluoromethyl substituted catalyst (**P4**) performed better than the 3,5-bistrifluoromethyl substituted catalyst (**P5**) (20% ee vs. 3% ee). These results suggested that these catalysts will operate in a manner that is different from that of **P1**–**P3**. To achieve high enantioselectivity, catalysts with bulkier substituents, such as 9-phenanthryl (**P6**), 9-anthracene (**P7**), and 2,4,6-isopropylphenyl (**P8**), were utilized. In these reactions, the greater the size of the substituent group was, the greater enantioselectivity was observed (**P6**, 55% ee; **P7**, 59% ee; **P8**, 69% ee). To our delight, higher enantioselectivity was achieved with the 2,4,6-tricyclohexyl-substituted catalyst (**P9**), which resulted in 86% yield and 82% ee of the desired product (Fig. 2, entry 1–9). When the VAPOL catalyst (**P10**) was employed, almost no enantioselectivity was observed (Fig. 2, entry 1–10). Using the most effective catalyst (**P9**), we screened several solvents (See the Supplementary Information for details) and determined that toluene was the optimal solvent in terms of yield and enantioselectivity (99% yield and 89% ee). While excellent enantioselectivity (95% ee) was observed at lower temperatures (−5 °C), chemical yield decreased to 74% (Fig. 2, entry 1–12). Fortunately, we observed that the reduced yield was compensated for by the addition

of 3 Å molecular sieves, resulting in 99% yield and 96% ee of the required product (Fig. 2, entry 1–13).

After establishing the optimal reaction conditions for *N*-arylindoles, we turned our attention to 2-arylindoles. The reaction of the regioisomeric 2-arylindole (**4a**) was performed using the previously established optimal reaction conditions that quickly yielded the desired product (**5aa**) in 99% yield and 66% ee (Fig. 2, entry 2–1). Due to the increased reactivity of the *C*−3 position compared to the *C*−2 position, the reaction occurred even in the absence of a catalyst (18% yield). To decrease the background rate, a lower concentration of **5aa** was shown to be beneficial, displaying 72% ee (Fig. 2, entry 2–2). In addition, a lower temperature (−20 °C) resulted in greater enantioselectivity (81% ee) despite the longer reaction time (Fig. 2, entry 2–3). The highest enantioselectivity for 2-arylindole was achieved at −78 °C, providing **5aa** in 99% yield and 83% ee (Fig. 2, entry 2–5).

As 3-arylindole is the reversed form of *N*-arylindole, we assumed that 3-arylindole would also exhibit a high degree of selectivity. In fact, when we performed the reaction of 3-arylindole (**6a**) under optimal conditions for *N*-arylindoles, we observed an enantioselectivity of 86% ee (Fig. 2, entry 3–1). However, in this reaction, the chemical yield drastically decreased, yielding only 7% of the required product (**7aa**). To increase the reaction rate while maintaining enantioselectivity, the reaction parameters were evaluated. When Lewis acids were employed to accelerates the reaction rate, slightly higher yields but lower enantioselectivities were observed (See the Supplementary Information for details). The reaction in dichloromethane rather than toluene produced a slightly better yield (Fig. 2, entry 3–2). As the optimal concentration corresponded to the minimum solvent volume (0.2 mL) required to properly mix the reaction, a higher concentration could not be used. When the reaction was conducted at a higher temperature, enantioselectivity decreased significantly. Eventually, the excess quantity of **2a** was used, providing **7aa** in 39% yield and 89% ee (Fig. 2, entry 3–3). A further increase in catalyst loading increased the yield to

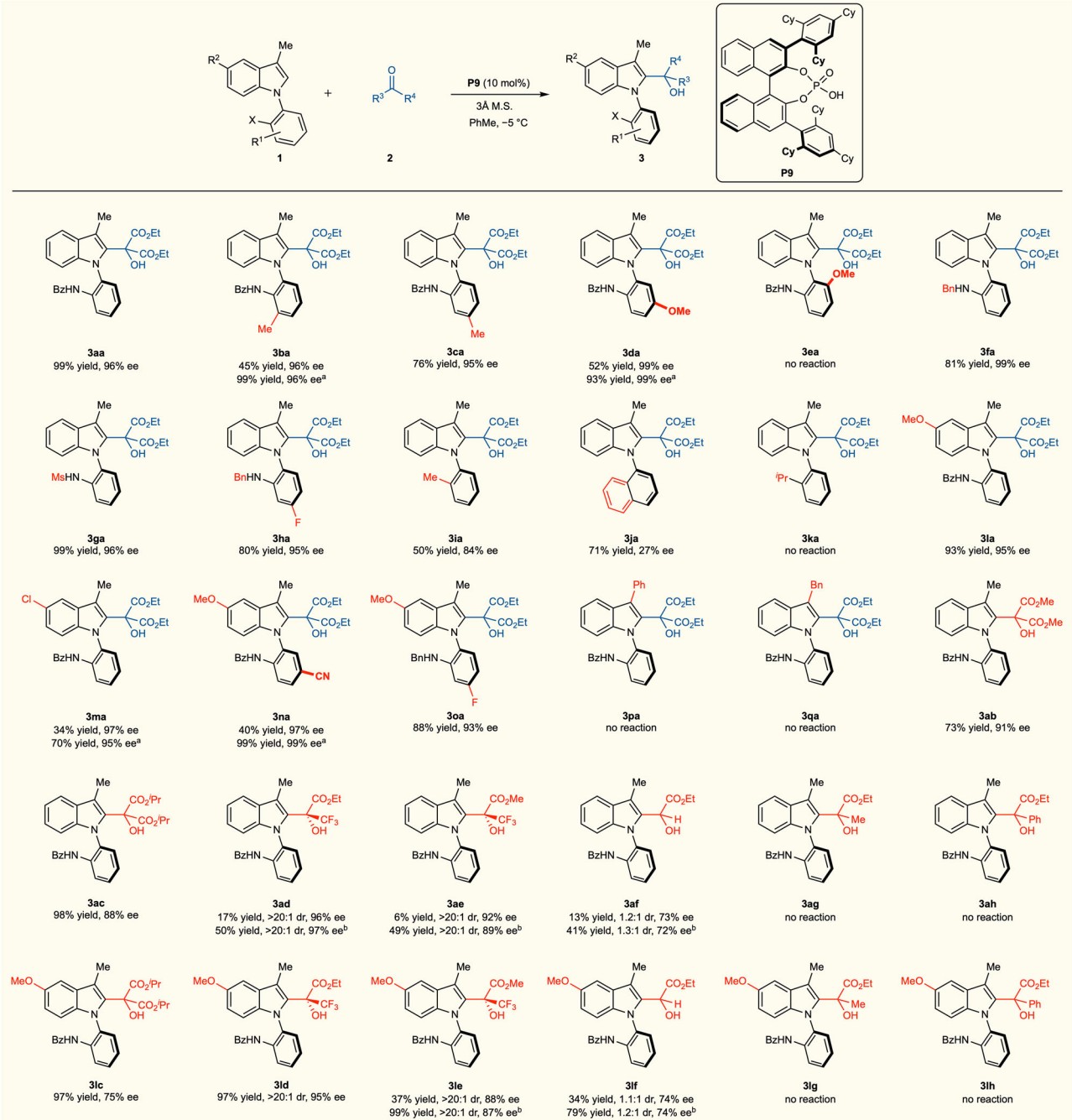

**Fig. 3 | Catalytic atroposelective synthesis of *N*-arylindoles.** Unless otherwise noted, the reactions were conducted with **1** (0.050 mmol, 1.0 equiv), **2** (0.075 mmol, 1.5 equiv), **P9** (0.005 mmol, 10 mol%, 5.0 mg), and 3 Å molecular sieves (40 mg) in PhMe (0.2 mL) at −5 °C. [a]20 mol% of **P9** was employed. [b]The reaction was performed at room temperature with 20 mol% of **P9**.

65% while retaining enantioselectivity (Fig. 2, entry 3–5). We believe that these optimizations reflect the different reactivities of the *C*−2 and *C*−3 positions in the indole and a subtle distinction between substrates in asymmetric catalysis.

### Evaluation of the substrate scopes

Using the optimal reaction conditions for *N*-arylindoles, 2-arylindoles, and 3-arylindoles, the substrate scopes were explored. Initially, substituted *N*-arylindoles were investigated, which demonstrated the broad applicability of our methodology (Fig. 3). When the bottom aromatic ring was substituted with a methyl group at the *ortho* position of the NHBz group, excellent enantioselectivity was observed;

however, the chemical yield was low (45% yield, 96% ee). To address this issue, 20 mol% of **P9** was utilized, resulting in 99% yield and 96% ee for **3ba**. The methyl substitution at the *para* position relative to the stereogenic axis was found to be acceptable and resulted in excellent enantioselectivity (76% yield, 95% ee). The reaction of the methoxy-substituted substrate at the *para* position of NHBz (**1d**) produced the intended product (**3da**) with even higher enantioselectivity but with a moderate yield (52% yield, 99% ee). The chemical yield was easily enhanced with 20 mol% of **P9** while maintaining enantioselectivity (93% yield, 99% ee). However, the substrate (**1e**) bearing a methoxy group in the *ortho* position of the stereogenic axis did not activate the reaction under the optimal reaction conditions, likely due to the steric

repulsion between the methoxy group and **2a**. The replacement of a benzamide group with a benzylamino group and a methanesulfonamide group produced the alcohol products **3fa** (81% yield and 99% ee) and **3ga** (99% yield and 96% ee). Accordingly, the reaction of **3h** with a benzylamino group and a fluoride group produced a high atroposelectivity product (80% yield and 95% ee).

A benzamide group was replaced with either an alkyl or aryl group to achieve high enantioselectivity. In fact, the methyl-substituted substrate (**1i**) demonstrated a considerable degree of enantioselectivity (84% ee). The *N*-naphthylindole (**1j**) reaction produced the intended product in 71% yield and 27% ee. However, the substrate (**1k**) bearing an isopropyl group did not affect the reaction. While the modification of the indole ring was comparable with our methodology, the reactivity varied depending on the substituent. For example, 5-methoxy-*N*-arylindole (**1l**) and 5-chloro-*N*-arylindole (**1m**) exhibited significantly different reactivity but comparable enantioselectivity (**3la**, 93% yield, 95% ee; **3ma**, 34% yield, 97% ee). The chemical yield of **3ma** was increased to 70% by utilizing 20 mol% of **P9**. Further substitutions of **1l** with electron-withdrawing groups, such as cyano and fluoride groups, were tolerated even though 20 mol% of **P9** was necessary to obtain a high yield of **3na**. The methyl group at the *C*−3 position was substituted with a phenyl group or a benzyl group (**1p** and **1q**), which did not yield the desired products. This incompatibility can be attributed to the steric hindrance between the catalyst and the *C*−3 position of indole in the transition states. When the reaction was performed with dimethyl ketomalonate (**2b**) or diisopropyl ketomalonate (**2c**) instead of diethyl ketomalonate (**2a**), the expected products (**3ab** and **3ac**) were obtained in 73% yield, 91% ee and 98% yield, 88% ee, respectively. As unsymmetric electrophiles might produce and regulate the stereogenic axis and quaternary stereogenic center, ethyl trifluoropyruvate (**2d**) was used as an electrophile for the catalytic reaction. Even though the reaction was slow under optimal conditions, excellent enantio- and diastereoselectivities were observed (17% yield, >20:1 dr, and 96% ee). When the reaction was performed with 20 mol% of **P9** at room temperature, the desired product (**3ad**) was obtained in 50% yield, >20:1 dr, and 97% ee. The reaction of **1a** with methyl trifluoropyruvate (**2e**) yielded the desired product (**3ae**) in 6% yield, >20:1 dr, 92% ee under the optimal reaction conditions, and 49% yield, >20:1 dr, 89% ee with 20 mol% of **P9** at room temperature. While methyl 2-oxoacetate (**2f**) produced the desired product albeit low selectivity (~1:1 dr, 73% ee), methyl 2-oxopropanoate (**2g**) and methyl 2-oxo-2-phenylacetate (**2h**) were found to be incompatible with our methodology. When the substrate with a 5-methoxy substitution (**1l**) was employed instead of **1a**, the reaction rate increased, but the selectivity was either maintained or slightly lower. The reaction of 5-methoxy-*N*-arylindole (**1l**) with **2c** produced **3lc** with moderate enantioselectivity. The reaction of **1l** with **2d**, **2e**, and **2f** provided the desired product in higher yields and similar selectivities under the optimized reaction conditions. However, **2g** and **2h** still did not yield the desired product.

Substrate scopes for 2-arylindoles were then examined with the optimized reaction conditions, which overall exhibited high enantioselectivities (Fig. 4, left). When the methoxy group is introduced at the *para* position of the benzamide, excellent yield and selectivity were observed (**5ba**, 95% yield, 96% ee). The replacements of the benzamide group with 4-methylbenzamide and 4-bromobenzamide provided the desired products **5ca** and **5da** (**5ca**, 99% yield, 87% ee; **5da**, 99% yield, 82% ee). The substrate (**4e**) containing a benzylamino group instead of a benzamide group was reacted with **2a** to produce the desired product in 99% yield and 90% ee. When the benzamide group was substituted with an alkyl or aryl group, 2-arylindoles produced superior results than *N*-arylindoles (**3ia**, **3ja**, and **3ka** vs. **5fa**, **5ga**, and **5ha**). For example, the reaction of a substrate containing a methyl group produced **5fa** in 96% yield and 87% ee. The catalyst induced the reaction of 2-naphthylindole (**4g**) to produce **5ga** in 40% yield and 77% ee. *N*-arylindole bearing an isopropyl group did not react with **2a**; however,

2-arylindole substrate (**4h**) and **2a** provided the desired product (**5ha**) with 59% yield and 43% ee, owing to the strong reactivity of the *C*−3 position. However, the reaction of the substrate containing a phenyl group (**4i**) failed to produce the desired product. 5-methoxy-2-arylindole (**4j**) interacted with **2a** in the presence of **P9** to produce 99% yield and 94% ee. Interestingly, a distinct tendency was observed compared to *N*-arylindoles when **2b** and **2c** were employed. The reaction between compound **4a** and dimethyl ketomalonate (**2b**) produced the atropisomeric product with very low enantioselectivity (14% ee). However, diisopropyl ketomalonate (**2c**) exhibited an even higher degree of enantioselectivity than diethyl ketomalonate (**2b**) (96% ee vs. 83% ee), implying that the origin of enantioselectivity for 2-arylindoles would be markedly distinct from that of *N*-arylindoles. When the methyl group at the *N*−1 position was substituted with an isopropyl group (**4k**), the selectivity reduced significantly to 44% ee. The reaction of **4a** with unsymmetrical ketones (**1d** and **1e**) resulted in the desired product with excellent yield, but low selectivity.

Then, we investigated 3-arylindoles under optimal reaction conditions (Fig. 4, right). The addition of a methyl group to the *para* position of the stereogenic axis was tolerated (**7ba**, 87% yield, 87% ee). When the benzamide group was substituted with a methyl group (**6c**), yield and selectivity decreased, further emphasizing the importance of the benzamide group. Interestingly, substituting the 5-position of indole with a methoxy group increased reactivity as the catalyst induced the catalytic reaction gradually under the optimal reaction conditions for *N*-arylindole (**7da**, 99% yield, 86% ee; **7ea**, 72% yield, 80% ee). While the substrates in which the methyl group at the *N*−1 position was substituted with an ethyl group (**6f**) or propyl group (**6g**) produced the desired products, the enantioselectivities were significantly reduced (40% ee for **7fa** and 22% ee for **7ga**). Substitution with a benzyl group (**6h**) or the removal of the substitution (**6l**) failed to yield the desired product. The reaction of 5-methoxy-3-arylindole (**6d**) with diisopropyl ketomalonate (**2c**) resulted in the desired product with 80% yield and 79% ee. The unsymmetrical electrophiles (**2d** and **2e**) were employed for the reaction of **6d**, which led to excellent yields and good to moderate enantioselectivities (>20:1 dr, 86% ee for **7dd** and 9:1 dr, 77% ee for **7de**).

## Computational studies and mechanistic consideration

To investigate the configurational stability of reaction products, the rotational barriers of **3aa**, **5aa**, and **7aa** were calculated, as shown in Fig. 5a. According to density functional theory (DFT) calculations (See the Supplementary Information for details)[62], **3aa** and **7aa** would have similarly high rotational barriers (36.7 and 36.6 kcal/mol, respectively), whereas **5aa** would have the highest rotational barrier (39.1 kcal/mol). These results indicated that the products would be configurationally stable and widely applicable in various areas without modification. The absolute stereochemistry was then determined by obtaining single-crystal X-ray structures of **3ba**, **3ld**, and **7da**, which were assigned as axial *R*-configured (Fig. 5b). To clarify the situation and assign the absolute stereochemistry, a chiroptical method was employed[63,64]. The experimental electronic circular dichroism (ECD) spectra of **3aa**, **5aa**, and **7aa** were measured and compared with their respective time-dependent DFT-calculated ECD spectra (See the Supplementary Information for details)[62], as depicted in Fig. 5c. The measured ECD spectra of **3aa** and **7aa** well-matched the calculated ECD spectrum of axial *R*-configured molecules. These results demonstrated that our ECD calculations were in good agreement with the single-crystal X-ray structures. Similarly, the absolute stereochemistry of **5aa** was determined to be axially *S*-configured.

To investigate the source of the observed enantioselectivity, computational studies were conducted (See the Supplementary Information for details) (Fig. 5d)[62]. Because of the voluminous substituents of chiral phosphoric acid (**P9**), the two-layer quantum mechanical (QM)/semiempirical (SE) ONIOM model was used[65–67]. The phosphoric acid group [−O$_2$P(O)OH] in **P9** and substrates were

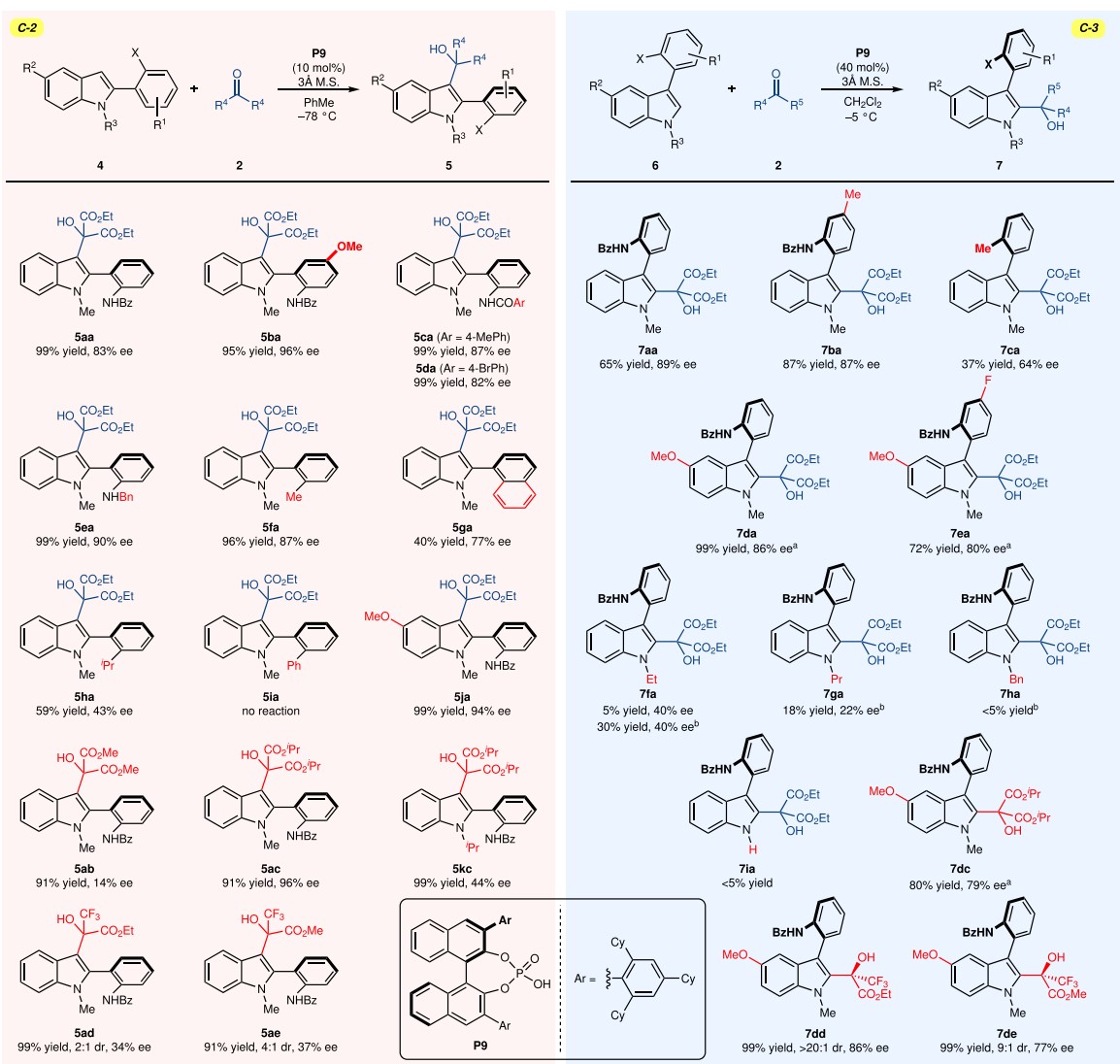

**Fig. 4 | Catalytic atroposelective synthesis of 2- and 3-arylindoles.** The reactions for **5** were conducted with **4** (0.050 mmol, 1.0 equiv), **2** (0.075 mmol, 1.5 equiv), **P9** (0.005 mmol, 10 mol%, 5.0 mg), and 3 Å molecular sieves (400 mg) in PhMe (2.0 mL) at −78 °C. Unless otherwise noted, the reactions for **7** were conducted with **6** (0.050 mmol, 1.0 equiv), **2** (1.125 mmol, 15 equiv), **P9** (0.020 mmol, 40 mol%, 20.0 mg), and 3 Å molecular sieves (40 mg) in PhMe (0.2 mL) at −5 °C. [a]The reaction was performed in PhMe with 10 mol% of **P9**.

assigned to the QM layer while the remaining atoms of **P9** were coated with the SE layer[68]. The QM layer was treated with M06-2X/6-31 G(d) while the SE layer was treated with PM6 for the geometry optimizations. Single-point energies of these optimized structures were calculated using M06-2X/def2-TZVP for the QM layer and PM6 for the SE layer with the inclusion of solvation energy corrections (SMD = toluene for **3aa** and **5aa**; SMD = dichloromethane for **7aa**). The obtained Gibbs free energies were corrected by zero-point vibrational energy (ZPVE) and temperature (268.15 K for **3aa** and **7aa** and 195.15 K for **5aa**). The corrected Gibbs free energy of the transition state (**TS1**) of **3aa** was calculated to be 2.06 kcal/mol lower than that of the transition state (**TS2**) of the enantiomer of **3aa**, which corresponds to our experimental result (Fig. 5d, top). Noncovalent interactions (NCI) analysis (See the Supplementary Information for details)[69] of **TS2-3aa** revealed unfavorable interactions between the substituents of **P9** and the benzyl group of **1a**. Similar transition state structures (**TS1-7aa** and **TS2-7aa**) to **TS1-3aa** and **TS2-3aa** were identified for **7aa**, with **TS1-7aa** being 1.47 kcal/mol less energetic than **TS2-7aa** (Fig. 5d, bottom). The existing transition structures indicated steric interactions between the catalyst and the benzoyl group of **6a**. For 2-arylindoles, the transition

state structures for **5aa** and the enantiomer of **5aa** are shown in Fig. 5d (middle). Although the energy gap between **TS1** and **TS2** for **5aa** (1.04 kcal/mol) was smaller than for **3aa** or **7aa**, it was in good agreement with our experimental result. Based on our NCI analysis, **TS2-5aa** exhibited unfavorable steric interactions between the catalyst and the benzoyl group of **4a**. As illustrated in **TS1-3aa** and **TS1-7aa**, the methyl group at the N−1 position of **3aa** and the C−3 position of **7aa** are directed toward the voluminous substituent of **P9**, resulting in limited space between the methyl group and the catalyst. This restricted space might explain the observed low reactivity in **3pa, 3qa**, and **7ha**.

## Biological evaluation

Given the biological application of indole scaffolds[41–43], particularly as anticancer drugs[70,71], the antiproliferative properties of **3, 5**, and **7** were examined using the SRB assay (Fig. 6). To avoid any biological bias for enantiopure compounds, early tests were conducted on selected racemic mixtures. Interestingly, independent of the position of an aromatic ring, several compounds exhibit significant antiproliferative effects with an IC$_{50}$ value of 10−20 μM against numerous cancer cell lines. For example, among N-arylindoles (Fig. 6a, left), (±)-**3ma** bearing

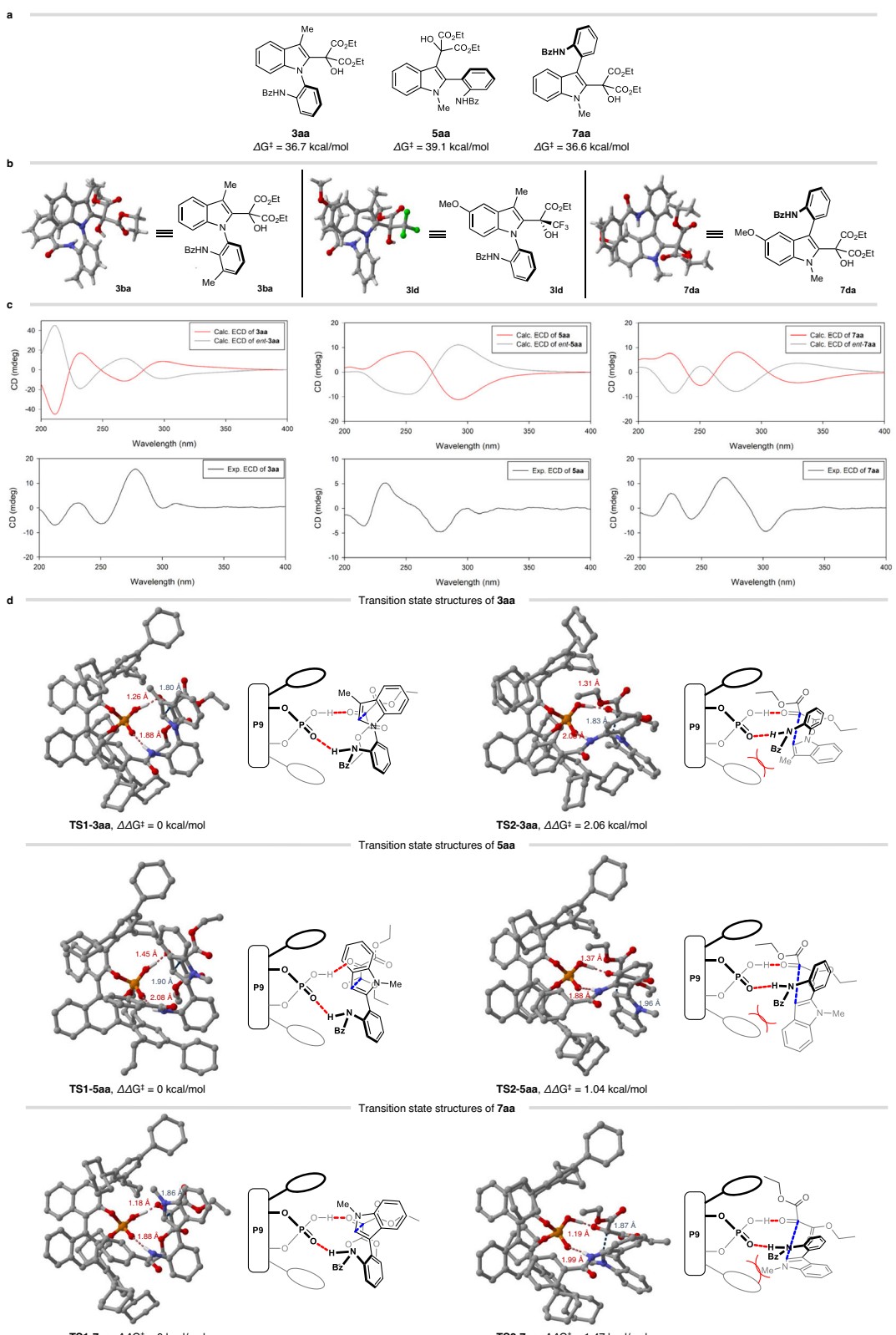

**Fig. 5 | Computational studies and mechanistic consideration. a** Calculated rotational barriers for **3aa**, **5aa**, and **7aa**. **b** X-ray crystal structures of **3ba**, **3ld**, and **7da**. **c** Calculated and experimental ECD spectra of **3aa**, **5aa**, and **7aa**. **d** Transition state structures for synthesizing **3aa**, **5aa**, and **7aa**. (All H atoms are omitted for clarity).

a chloro group at the $C-5$ of indole exhibited promising antiproliferative effects with an $IC_{50}$ of 13.4−20.8 µM. In addition, (±)-**5da** and (±)-**7ba** exhibited significant biological activities against five human cancer cell lines. This result shows that additional medicinal chemistry investigations will afford novel, promising anticancer drugs. Further biological studies of each enantiomer of **3ma** were performed, as shown in Fig. 6b. The antiproliferative activity of *ent*-**3ma**, which was synthesized in the presence of *ent*-**P9**, was overall greater than that of

**a**

| | A549 | HCT116 | MDA-MB-231 | SK-Hep-1 | SNU-638 |
|---|---|---|---|---|---|
| (±)-3aa | 20.6 ± 2.0 | 45.0 ±1.0 | 24.8 ± 3.6 | 21.5 ± 0.1 | 25.2 ± 0.1 |
| (±)-3ba | >50 | >50 | >50 | >50 | 42.7 ± 1.8 |
| (±)-3ca | 19.6 ± 2.3 | 34.6 ± 2.9 | 19.8 ±1.3 | 21.7 ±1.0 | 17.0 ± 2.5 |
| (±)-3da | 18.6 ± 2.4 | 45.6 ± 1.7 | 30.2 ± 0.7 | 25.8 ± 3.2 | 19.8 ± 1.0 |
| (±)-3fa | 19.0 ± 1.2 | 45.0 ± 2.1 | 30.0 ± 0.5 | 21.5 ± 2.6 | 21.2 ± 0.5 |
| (±)-3ga | >50 | >50 | >50 | >50 | >50 |
| (±)-3ha | 16.6 ± 1.3 | 30.6 ± 2.9 | 25.8 ± 1.4 | 22.6 ± 3.4 | 14.9 ± 0.6 |
| (±)-3ia | 24.2 ± 2.2 | >50 | 41.6 ± 3.0 | 44.0 ± 4.8 | 29.6 ± 2.3 |
| (±)-3la | 27.3 ± 3.2 | >50 | 39.9 ± 1.8 | 24.4 ± 5.5 | 30.7 ± 2.2 |
| (±)-3ma | 13.7 ± 1.1 | 20.8 ± 4.1 | 13..4 ± 1.1 | 15.2 ± 1.7 | 18.2 ± 0.3 |
| (±)-3na | 27.4 ± 0.1 | >50 | >50 | >50 | 32.99 ± 0.7 |
| (±)-3oa | 12.3 ± 0.6 | >50 | >50 | 24.2 ± 2.3 | 16.6 ± 1.0 |
| (±)-3ab | >50 | >50 | >50 | >50 | >50 |
| (±)-3ac | 16.0 ± 0.4 | 40.2 ± 6.6 | 39.0 ± 3.7 | 26.4 ± 1.8 | 21.2 ± 0.7 |
| Irinotecan | 4.0 ± 0.8 | 7.1 ± 2.4 | 9.1 ± 2.1 | 4.4 ± 1.3 | 1.7 ± 0.1 |

| $IC_{50}{}^a$ (µM) | | | | | |
|---|---|---|---|---|---|
| | A549 | HCT116 | MDA-MB-231 | SK-Hep-1 | SNU-638 |
| (±)-5aa | 46.64 ± 1.4 | >50 | >50 | >50 | 34.0 ± 0.9 |
| (±)-5ba | 45.6 ± 2.1 | >50 | >50 | 27.5 ± 0.3 | 19.0 ± 2.3 |
| (±)-5ca | 36.3 ± 7.3 | 41.0 ± 2.3 | 27.6 ± 4.9 | 20.7 ± 1.5 | 21.2 ± 0.8 |
| (±)-5da | 17.4 ± 0.8 | 30.9 ± 3.1 | 21.0 ± 0.1 | 20.3 ± 4.2 | 15.3 ± 1.7 |
| (±)-5ja | >50 | >50 | >50 | >50 | 38.5 ± 4.7 |
| (±)-5ab | >50 | >50 | >50 | >50 | >50 |
| (±)-5ac | 24.2 ± 0.3 | 43.4 ± 1.2 | 35.7 ± 0.2 | 25.9 ± 2.1 | 25.4 ± 6.1 |
| Irinotecan | 3.6 ± 1.2 | 6.9 ± 2.5 | 7.8 ± 0.8 | 4.4 ± 0.2 | 1.7 ± 0.2 |

| | A549 | HCT116 | MDA-MB-231 | SK-Hep-1 | SNU-638 |
|---|---|---|---|---|---|
| (±)-7aa | 17.9 ± 0.1 | 23.1 ± 3.2 | 22.6 ± 1.4 | 21.6 ± 3.9 | 24.8 ± 1.4 |
| (±)-7ba | 13.3 ± 0.1 | 15.2 ± 2.4 | 12.0 ± 0.5 | 15.3 ± 3.9 | 15.3 ± 0.3 |
| (±)-7ca | 13.0 ± 1.6 | 32.0 ± 7.5 | 13.1 ± 0.4 | 12.3 ± 0.6 | 15.4 ± 1.1 |
| (±)-7da | 29.9 ± 5.5 | >50 | >50 | >50 | 35.6 ± 2.6 |
| (±)-7ea | >50 | >50 | >50 | >50 | >50 |
| (±)-7dc | 15.4 ± 6.3 | 42.0 ± 4.8 | 41.8 ± 4.5 | 34.4 ± 5.0 | 30.0 ± 2.9 |
| Irinotecan | 3.9 ± 0.4 | 6.0 ± 2.1 | 6.4 ± 1.1 | 4.1 ± 1.0 | 1.4 ± 0.6 |

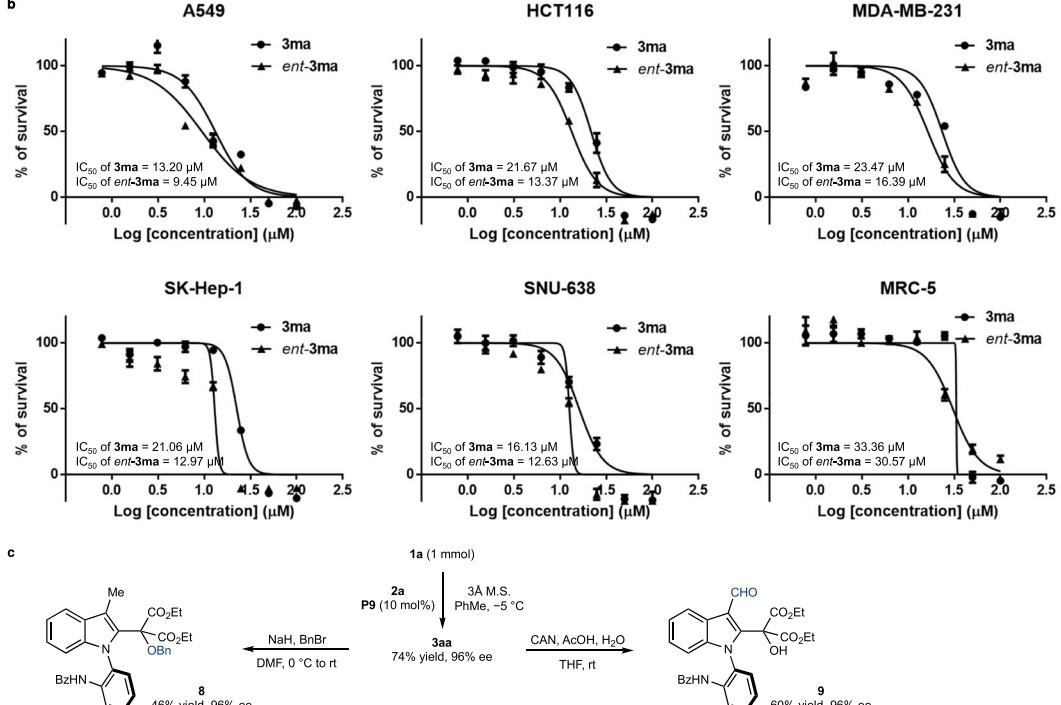

**Fig. 6 | Antiproliferative activities of selected atropisomeric indoles and further chemical modifications. a** Antiproliferative activities of the racemic mixtures of selected atropisomeric indoles. $^a$All values are the means of at least three experiments. Measured by the Sulforhodamine-B (SRB) method. Data are represented as means ± SEM. **b** Antiproliferative activities of **3ma** and *ent*-**3ma**. The results are representative of three independent experiments. Measured by the SRB method. Data are represented as means ± SD. **c** One-mmol-scale reaction and chemical derivatization of product. A549: Human lung cancer cell line; HCT116: Human colon cancer cell line; MDA-MB-231: Human breast cancer cell line; SK-Hep-1: Human liver cancer cell line; SNU-638: Human stomach cancer cell line; MRC-5: Normal human lung fibroblast cell line.

**3ma.** When both atropisomers were treated with normal human lung fibroblast cell lines to test toxicity, they exhibited approximately 2- to 3-fold lower antiproliferative activities compared to cancer cells. Although the variation in antiproliferative activity between cancer and normal cell lines was not dramatically significant, this study highlighted the significance of atroposelective synthesis of a biologically relevant scaffold and dependable atroposelective strategy[72–74]. To verify our results, the compound **3aa** was exposed to Dulbecco's Modified Eagle Medium (DMEM) medium for 72 h at 37 °C, in which any decomposition including hydrolysis was not observed. To further demonstrate the applicability of our methodology, a 1-mmol-scale

reaction, and derivatizations were conducted (Fig. 6c). The reaction of 1 mmol of **1a** with **2a** in the presence of **P9** resulted in the formation of the desired product in 74% yield and 96% ee. Subsequently, the product **3aa** underwent benzylation to yield compound **8** without any loss of enantioselectivity. In addition, the oxidation of **3aa** provided indole 3-carboxaldehyde compound (**9**), which can be further transformed into diverse compounds.

## Discussion

Chemists have made tremendous efforts to increase the availability of chiral compounds using asymmetric catalysis, which has led to the

development of invaluable synthetic methods. To move on to the subsequent phase of asymmetric catalysis, we believe that one of the characteristics of a future synthetic methodology would be its high generality. It would be highly desirable that the scope covers a related but distinct type of substrates as well as a variety of substitution patterns. This objective would be achieved by developing a new strategy informed by valuable precedents. In this regard, we would like to propose a general strategy for atroposelective dynamic kinetic resolution using *o*-amidobiaryls, as evidenced by the asymmetric synthesis of different types of arylindoles. In the presence of chiral phosphoric acid, three types of arylindoles containing an aromatic ring in the *N*−1, *C*−2, or *C*−3 position exhibit high yields and high enantioselectivities (up to 99% ee). Our concept has been further demonstrated by computational studies, in which hydrogen bonding plays a crucial role in achieving high atroposelectivities. In addition, a biological evaluation of our unique scaffold reveals potent anti-proliferative capabilities, allowing for the future development of an anticancer drug. We believe that our general approach might be extended to more substrate types. Furthermore, we expect that this study will serve as a stepping stone to develop a new approach for the next highest level of generalizability.

## Methods

### Representative procedure for the catalytic atroposelective functionalization

To an oven-dried 4 mL vial equipped with a magnetic stir bar was added substrate **1** (0.05 mmol, 1 equiv), **P9** (0.005 mmol, 0.1 equiv), and 3 Å molecular sieves (40 mg). Then, the solution of **2** (0.075 mmol, 1.5 equiv) in PhMe (0.2 mL, 0.25 M) was added. The vial was sealed with a Teflon cap and further secured with Parafilm M®. The reaction mixture was stirred at −5 °C until complete consumption of **1**. Then the solvent was removed in vacuo. The crude material was purified by flash column chromatography using an eluent of 5% EtOAc/CH$_2$Cl$_2$ to afford the desired material **3**. The enantioselectivity was determined by chiral HPLC.

### Reporting summary

Further information on research design is available in the Nature Portfolio Reporting Summary linked to this article.

## Data availability

The X-ray crystallographic coordinates for structures reported in this study have been deposited at the Cambridge Crystallographic Data Centre (CCDC), under deposition numbers CCDC 2219791 (**3ba**), 2284844 (**3ld**), and 2219782 (**7da**). These data can be obtained free of charge from The Cambridge Crystallographic Data Centre via http://www.ccdc.cam.ac.uk/data_request/cif. Detailed experimental procedures and characterization of compounds as well as NMR and HPLC spectra can be found in the Supplementary Information (PDF). The coordinates of the optimized structures are provided in the Source Data file (XLS).  Correspondence and requests for materials should be addressed to Y.K. Source data are provided with this paper.

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

## Acknowledgements

This work is supported by a National Research Foundation of Korea (NRF) grant funded by the Korean government (MSIT) (2020R1C1C1006231, 2022R1A4A1018930, and 2022M3E5F2017857 for Y.K.). This work is also supported by the National Supercomputing Center with supercomputing resources including technical support (KSC-2022-CRE-0288 for Y.K.) and the POSCO Science Fellowship of POSCO TJ Park Foundation (Y.K.).

## Author contributions

A.K. carried out the synthesis of the catalysts, optimization of the reaction conditions, and most of the substrate scope. She also conducted computational studies and wrote the supplementary information. C.L. contributed to the optimization of the reaction conditions and investigation of the substrate scope. J.S. conducted the biological experiments. S.K.L. designed the biological experiments. Y.K. designed and supervised the project and wrote the paper with contributions from all authors.

## Competing interests

The authors declare no competing interests.
