## [Peer Review File · Nature Communications]

All-round catalytic and atroposelective strategy via dynamic kinetic resolution for N-/2-/3-arylindolesReviewers' Comments:

Reviewer #1:

Remarks to the Author:

Kwon and co-workers describe in this manuscript atroposelective construction of N-2 and N-3 arylindoles using chiral phosphoric acid. This is an excellent extension of the previous work of the group (ref. 54). Wide scope of substrates is demonstrated and mechanistic study and biological activity was evaluated.

Construction of C-N axial chirality stereoselectively has attracted much attention of organic chemists and a number of excellent methods have been reported lately. This manuscript will pave the way for the C-N axial chirality formation.

This reviewer recommends publication of the manuscript in Nat. Commun. after addressing following issues.

(1) Ref. 5, it is recommended to update.

Akiyama, T. Ojima, I. *Catalytic Asymmetric Synthesis*, Fourth Edition, Jon Wiley & Sons (2022).

(2) Please add following review articles.

Rodríguez-Salamanca, P.; Fernández, R.; Hornillos, V.; Lassaletta, J. M. *Asymmetric Synthesis of Axially Chiral C–N Atropisomers*. *Chem. Eur. J.* 2022, 28, e202104442.

Pellissier, H. *Organocatalytic Dynamic Kinetic Resolution: An Update*. *Eur. J. Org. Chem.* 2022, 2022, e202101561.

(3) Following recent articles should be included.

Choi, S.; Guo, M. C.; Coombs, G. M.; Miller, S. J. *Catalytic Asymmetric Synthesis of Atropisomeric N-Aryl 1,2,4-Triazoles*. *J. Org. Chem.* 2023. DOI: 10.1021/acs.joc.2c02727.

Reviewer #2:

Remarks to the Author:

This manuscript reports a chiral phosphoric acid-catalyzed atroposelective synthesis of N-arylindoles, 2-arylindoles and 3-arylindoles through dynamic kinetic resolution strategy. The authors designed o-amidobiaryl substrates for the interaction with CPA catalyst. As we have seen, this method features broad substrate tolerance of arylindoles and ketomalonate, reasonable structure outcome by DFT calculation and positive biological study. In terms of novelty of the reaction, this reviewer considers that this manuscript is suitable for its publication in Nature Communications after addressing the following issues.

1. The catalyst loading for the atroposelective synthesis of 3-arylindole 7aa is rather high. Did the authors try to add some Lewis acids for the activation of ketomalonate?

2. In Fig 3, the authors examined N-arylindole 1 bearing C3-methyl group. What about other groups, such as phenyl, benzyl at C3 carbon?

3.) If the N-unprotected 2-arylindole reacted with ketomalonate under the optimal conditions, what would happen?

4. Is there an explainable reason for the trace yield of 7fa when benzyl group instead of methyl group at N-position?

5. How to figure out the absolute structure of 7dd?

Reviewer #3:

Remarks to the Author:

In this work Kwon and coworkers present a catalytic strategy for the atroposelective dynamic kinetic resolution of arylindoles. The proposed approach appears especially successful for N-arylindoles, with excellent ee for a broad range of substrates. For 2- and 3- arylindoles the reaction scope is significantly more limited and "excellent" selectivities were obtained only in a handful of cases.

Overall, this is a nice study that deserves to be published somewhere. However, considering the limited reaction scope, I do not believe that it has the impact that is required from this journal. In addition, I have some technical remarks related to the computational part that the authors should address before the manuscript can be considered for publication in any journal.

1) In computing relative energies between transition states, the authors write that the observed variations are consistent with the experimental results. This is not entirely correct because energy differences of 4-5 kcal/mol would correlate with much higher selectivities. This aspect should be discussed.

2) Different reaction conditions were used for N-2-/3-arylindoles. However, it appears that the same computational settings were used in all situations. This aspect should be discussed, with particular regard to temperature effects.

3) For these systems noncovalent interactions are extremely important, as also emphasized by the authors several times in the text. However, the authors did not use dispersion corrections in the B3LYP part of their geometry optimizations. While single point M062X calculations are probably adequate, the level of theory used for the geometries requires a justification.

4) The conformational sampling procedure used for the transition states should be discussed in more detail. The final energy ordering was based on B3LYP/DZ or M062X/TZ? This is crucial because B3LYP/DZ energies without dispersion corrections are most likely not accurate enough for relative energy estimates.

Reviewer #4:

Remarks to the Author:

In this manuscript, Kim et al. report a novel methodology using *o*-aminobiaryl as a flexible platform for atroposelective dynamic kinetic resolution using chiral phosphoric acids as the catalysts. N-, 2- and 3-arylindoles were reacted with ketomalonates in the presence of chiral phosphoric acids, and the 2,4,6-tricyclohexyl-substituted catalyst P9 provided the highest enantioselectivity with good yields. The mechanism for enantioselectivity was investigated using computational methods. Preliminary antiproliferative activity was studied to show the biological significance of the synthesized compounds. The work presented have several major issues that need to be addressed:

1. For the methodology study: (a) Broader applicability needs to be demonstrated for the work to qualify for publication in Nat Comm. For example, have the authors studied other electrophiles besides the ketomalonates? (b) For the asymmetric ketomalonates used in the work, the conformation of the newly formed chiral center should be demonstrated (for example compounds 3ad, 3id and 7dd). Also in this regard, more asymmetric ketomalonates should be studied. (c) Have the authors tried other N substituent for the substrate 6 (Figure 4)? N-Benzyl and NH showed very poor results. How about N-ethyl and the N-propyl groups? (d) There seems to be very limited space for medicinal chemistry modifications of the products generated by this method. A number of examples should be added to showcase such modifications.

2. For the antiproliferative activity study: (a) The authors used only four concentration points to produce the curves. This would lead to inaccuracy. At least eight points should be tested. (b) The compounds showed only moderate activity with IC₅₀ values in the range of 10-20 μM. Such effects should not be claimed as "great efficacy" in the abstract. Regarding the antiproliferative activity, have the authors tested the compounds in normal mammalian cells to measure toxicity? Specificity needs to be demonstrated for the anticancer potential of the compounds. (3) Compound stability should also be examined. For example, will the ester groups be hydrolyzed under the incubation conditions?

First and foremost, I am deeply grateful for the efforts to the reviewers have made to our manuscript. Our manuscript has greatly improved to the reviewer's feedback and insights. Our specific changes in response to the reviewer points follow below:

With Respect to Reviewer 1:

Reviewer 1: *Kwon and co-workers describe in this manuscript atroposelective construction of N-2 and N-3 arylindoles using chiral phosphoric acid. This is an excellent extension of the previous work of the group (ref. 54). Wide scope of substrates is demonstrated and mechanistic study and biological activity was evaluated. Construction of C-N axial chirality stereoselectively has attracted much attention of organic chemists and a number of excellent methods have been reported lately. This manuscript will pave the way for the C-N axial chirality formation. This reviewer recommends publication of the manuscript in Nat. Commun. after addressing following issues.*

Our response: We greatly appreciate the positive assessment.

Reviewer 1 Point 1: *Ref. 5, it is recommended to update. Akiyama, T. Ojima, I. Catalytic Asymmetric Synthesis, Fourth Edition, Jon Wiley & Sons (2022).*

Our response: Of course, this reference has been updated.

Reviewer 1 Point 2: *Please add following review articles. Rodríguez-Salamanca, P.; Fernández, R.; Hornillos, V.; Lassaletta, J. M. Asymmetric Synthesis of Axially Chiral C–N Atropisomers. Chem. Eur. J. 2022, 28, e202104442. Pellissier, H. Organocatalytic Dynamic Kinetic Resolution: An Update. Eur. J. Org. Chem. 2022, 2022, e202101561.*

Our response: According to the comment, Fernández, Hornillos, and Lassaletta's *Chem. Eur. J.* paper has been added as reference 10, and Pellissier's *Eur. J. Org. Chem.* paper has been added as reference 34.

Reviewer 1 Point 3: *Following recent articles should be included. Choi, S.; Guo, M. C.; Coombs, G. M.; Miller, S. J. Catalytic Asymmetric Synthesis of Atropisomeric N-Aryl 1,2,4-Triazoles. J. Org. Chem. 2023. DOI: 10.1021/acs.joc.2c02727.*

Our response: According to the comment, the recent *J. Org. Chem.* paper has been added as reference 37.

With Respect to Reviewer 2:

Reviewer 2: *This manuscript reports a chiral phosphoric acid-catalyzed atroposelective synthesis of N-arylindoles, 2-arylindoles and 3-arylindoles through dynamic kinetic resolution strategy. The authors designed o-amidobiaryl substrates for the interaction with CPA catalyst. As we have seen, this method features broad substrate tolerance of arylindoles and ketomalonate, reasonable structure outcome by DFT calculation and positive biological study. In terms of novelty of the reaction, this reviewer considers that this manuscript is suitable for its publication in Nature Communications after addressing the following issues.*

Our response: We greatly appreciate the positive assessment.

Reviewer 2 Point 1: *The catalyst loading for the atroposelective synthesis of 3-arylindole 7aa is rather high. Did the authors try to add some Lewis acids for the activation of ketomalonate?*

Our response: As the reviewer suggested, we have tested several Lewis acid catalysts to activate ketomalonate with 20 mol% of **P9**.

The results are shown below.

Entry	Lewis Acid (x mol%)	Solvent	Yield ^a (%)	e.e. ^b (%)
1	None	PhMe	34	88
2	AlCl ₃ (10)	PhMe	29	86
3	CeCl ₃ (10)	PhMe	22	85
4	CuCl (10)	PhMe	7	85
5	CuCl ₂ (10)	PhMe	58	85
6	FeCl ₃ (10)	PhMe	37	77
7	InCl ₃ (10)	PhMe	48	75
8	MgCl ₂ (10)	PhMe	18	86
9	ZnCl ₂ (10)	PhMe	39	84
10	ZnCl ₂ (30)	PhMe	42	74

^aIsolated yields. ^bEnantiomeric excesses were determined by chiral HPLC analysis.

When several Lewis acids such as CuCl₂, FeCl₃, InCl₃, and ZnCl₂ were employed, the chemical yields were slightly improved. However, enantioselectivities decreased to a

degree, presumably due to the interference of interactions between the chiral phosphoric acid and substrate.

The results have been summarized in the Supplementary Information (Section 5.3.1), and the description has been added to the main text in Page 9. Now, it reads.

“...When Lewis acids were employed to accelerates the reaction rate, slightly higher yields but lower enantioselectivities were observed.⁶³”

Reviewer 2 Point 2: *In Fig 3, the authors examined N-arylindole 1 bearing C3-methyl group. What about other groups, such as phenyl, benzyl at C3 carbon?*

Our response: According to the reviewer comment, two new substrates containing a phenyl and benzyl group at the C-3 position were prepared and subjected to the catalytic reaction. In those reactions, unfortunately, the desired products were not observed under the optimized reaction conditions.

We believe that this is similar case to the Reviewer 2 Point 4. Because the aromatic ring is substituted at the N-1 position and ketomalonate approaches to the C-2 position, the C-3 position has no choice but to be headed to the bulky substituents of the catalyst in the transition state. Due to the steric hindrance, the large substituents are hard to accommodate at the C-3 position.

These results have been added to the main text in Page 10 and Fig. 3 (**3pa** and **3qa**). Now, it reads.

“...The methyl group at the C-3 position was substituted with a phenyl group or a benzyl group (**1p** and **1q**), which did not yield the desired products. This incompatibility can be attributed to the steric hindrance between the catalyst and the C-3 position of indole in the transition states.”

Reviewer 2 Point 3: *If the N-unprotected 2-arylindole reacted with ketomalonate under the optimal conditions, what would happen?*

Our response: This is interesting point. When the N-unprotected 2-arylindole reacted with ketomalonate under the optimal conditions, the C-3 substituted product was obtained. However, because of the absence of a substituent at the N-1 position, the rotational barrier around the stereogenic C-C bond does not seem to be high enough. For this reason, two atropisomeric products were not separated in our chiral HPLC analysis.

While further investigation on the 3-methyl-*N*-unprotected 2-arylimidoles has been made, desired product was not obtained in this reaction.

Even though an atropisomeric compound was not obtained in these reactions, the atroposelective synthesis of 2-arylimidole through catalytic substitution at the N-1 position is highly intriguing. We are currently underway to explore brand-new chemistry to control the C–C stereogenic axis through *N*-alkylation.

Reviewer 2 Point 4: *Is there an explainable reason for the trace yield of 7fa when benzyl group instead of methyl group at N-position?*

Our response: As shown in the computational study (Fig. 5d), the methyl group at the N-1 position is headed to the bulky tricyclohexylphenyl group of **P9**. The distance between the methyl group and the tricyclohexylphenyl group is about 3.12 Å in **TS1-7aa**. We believe that the benzyl group could not fit to this space and if it were to fit, it would weaken the interactions between the substrate and the catalyst, which would significantly slow down the reaction rate. This is the similar case to the **3pa** and **3qa**, which is raised by Reviewer 2 Point 2.

To clarify this issue, we have added some sentences to the main text in Page 17 to describe this issue. Now, it reads.

“...As illustrated in **TS1-3aa** and **TS1-7aa**, the methyl group at the N-1 position of **3aa** and the C-3 position of **7aa** are directed toward the voluminous substituent of **P9**, resulting in limited space between the methyl group and the catalyst. This restricted space might explain the observed low reactivity in **3pa**, **3qa**, and **7ha**.”

Reviewer 2 Point 5: *How to figure out the absolute structure of 7dd?*

Our response: To determine the absolute structure, we attempted to get single crystal of both **3ld** and **7dd**. Ultimately, the single crystal X-ray structure of **3ld** was obtained, which allowed us to determine its absolute configuration (CCDC Deposition Number: 2284844).

Because the compounds **3** and **7** are in a reversed form of each other, they exhibit similar transition state structures as shown in Fig. 5d. In this regard, we believe that they share the same pattern of absolute configuration. The supposed absolute configuration for **7dd** has been designated in Fig. 4.

Furthermore, we successfully grew single crystals of racemic mixtures of **7dd**, which allowed us the X-ray crystal structure of (\pm)-**7dd**, as shown below. It also supports the relative configuration between the stereogenic axis and stereogenic center, depicted in Fig. 4.

With Respect to Reviewer 3:

Reviewer 3: *This manuscript reports a chiral phosphoric acid-catalyzed atroposelective synthesis of N-arylindoles, 2-arylindoles and 3-arylindoles through dynamic kinetic resolution strategy. The authors designed o-amidobiaryl substrates for the interaction with CPA catalyst. As we have seen, this method features broad substrate tolerance of arylindoles and ketomalonate, reasonable structure outcome by DFT calculation and positive biological study. In terms of novelty of the reaction, this reviewer considers that this manuscript is suitable for its publication in Nature Communications after addressing the following issues.*

Our response: We understand the reviewer's concerns, but there are several points we would like to emphasize regarding our research:

- 1) We have explored challenging substrates to demonstrate the strengths and limitations of our methodology. This was done to clearly illustrate the scope of our research. If we had only made simple substitutions while retaining NHBz, we could have shown a much broader range of substrates. Especially, for 2-arylindoles, we have achieved up to 96% ee, and in most cases, we obtained high selectivities ranging from 80% to 96% ee, when the hydrogen-bond donor was present. In the case of 3-arylindoles, many examples exhibited high selectivities of over 80%, when hydrogen-bond donor was included.
- 2) We believe that these research findings can serve as a starting point for comprehensive discussions on the stereogenic axes that indoles can possess. In addition, the reaction products displayed excellent physiological activities without the need for additional modifications. This further emphasizes the significance of our research.

Reviewer 3 Point 1: *In computing relative energies between transition states, the authors write that the observed variations are consistent with the experimental results. This is not entirely correct because energy differences of 4-5 kcal/mol would correlate with much higher selectivities. This aspect should be discussed.*

Our response: We thought that the significant energy difference might have arisen due to the difference between B3LYP functional for the geometry optimization and M06-2X functional for the single point energy calculation in the context of the applied ONIOM model. To address this issue and other concerns raised by Reviewer 3, we performed computational calculations again, regarding to the transition state structures optimization using superior and more accurate M06-2X functional. As a result of this calculation, we were able to obtain energy difference results that fit more accurately (2.06 kcal/mol for **3aa**, 1.04 kcal/mol for **5aa**, and 1.47 kcal/mol for **7aa**).

We would like to express our gratitude to Reviewer 3 for providing valuable advice that allowed us to achieve improved calculation results. Accordingly, we have made appropriate revisions to the main text in Page 16 and Fig. 5. Now, it reads.

“...The corrected Gibbs free energy of the transition state (**TS1**) of **3aa** was calculated to be 2.06 kcal/mol lower than that of the transition state (**TS2**) of the enantiomer of **3aa**, which corresponds to our experimental result (Fig. 5d, top)... Similar transition state structures (**TS1-7aa** and **TS2-7aa**) to **TS1-3aa** and **TS2-3aa** were identified for **7aa**, with **TS1-7aa** being 1.47 kcal/mol less energetic than **TS2-7aa** (Fig. 5d, bottom)... Although the energy gap between **TS1** and **TS2** for **5aa** (1.04 kcal/mol) was smaller than for **3aa** or **7aa**, it was in good agreement with our experimental result.”

Reviewer 3 Point 2: Different reaction conditions were used for N-2-/3-arylindoles. However, it appears that the same computational settings were used in all situations. This aspect should be discussed, with particular regard to temperature effects.

Our response: According to the comment, we performed calculations again with different computational settings depending on the reaction conditions. As a result of these corrections, we were able to obtain results that better matched the experimental values.

The detailed computational settings have been described in the main text (Page 16) and Supplementary Information. Now, it reads.

“...Single-point energies of these optimized structures were calculated using M06-2X/def2-TZVP for the QM layer and PM6 for the SE layer with the inclusion of solvation energy corrections (SMD = toluene for **3aa** and **5aa**; SMD = dichloromethane for **7aa**). The obtained Gibbs free energies were corrected by zero-point vibrational energy (ZPVE) and temperature (268.15 K for **3aa** and **7aa** and 195.15 K for **5aa**).”

Reviewer 3 Point 3: For these systems noncovalent interactions are extremely important, as also emphasized by the authors several times in the text. However, the authors did not use dispersion corrections in the B3LYP part of their geometry optimizations. While single point M062X calculations are probably adequate, the level of theory used for the geometries requires a justification.

Our response: As pointed out by the Reviewer 3, we already used Grimme’s dispersion 3 to account for non-covalent interactions. However, unfortunately, we omitted this information to Supplementary Information, which caused confusion. We apologize for the oversight.

To clarify all concerns raised by Reviewer 3, we have performed computational calculations again using the superior M06-2X functional. We hope that with this revision, issues regarding the calculation results will be resolved. Accordingly, the main text (Page 16) and Supplementary Information have been revised. Now, it reads.

“...The QM layer was treated with M06-2X/6-31G(d) while the SE layer was treated with PM6 for the geometry optimizations.”

Reviewer 3 Point 4: The conformational sampling procedure used for the transition states should be discussed in more detail. The final energy ordering was based on B3LYP/DZ or M062X/TZ? This is crucial because B3LYP/DZ energies without dispersion corrections are most likely not accurate enough for relative energy estimates.

Our response: According to the comment, the detailed procedure for the conformational analysis has been added to the Supplementary Information in the Section 12.3.1.

In the revised version, the final energies were ordered by the Gibbs free energy computed using the QM/SE ONIOM model (M06-2X/6-31G(d):PM6) level of theory. With these modifications, we were able to obtain results that clearly incorporated dispersion corrections.

With Respect to Reviewer 4:

Reviewer 4: *In this manuscript, Kim et al. report a novel methodology using o-aminobiaryl as a flexible platform for atroposelective dynamic kinetic resolution using chiral phosphoric acids as the catalysts. N-, 2- and 3-arylindoles were reacted with ketomalonates in the presence of chiral phosphoric acids, and the 2,4,6-tricyclohexyl-substituted catalyst P9 provided the highest enantioselectivity with good yields. The mechanism for enantioselectivity was investigated using computational methods. Preliminary antiproliferative activity was studied to show the biological significance of the synthesized compounds. The work presented have several major issues that need to be addressed:*

Our response: We sincerely appreciate the detailed analysis of our manuscript, and we have diligently conducted additional research to address the issues raised by the reviewer.

Reviewer 4 Point 1: *For the methodology study: (a) Broader applicability needs to be demonstrated for the work to qualify for publication in Nat Comm. For example, have the authors studied other electrophiles besides the ketomalonates?*

Our response: Initially, we tested several electrophiles including *N*-chlorosuccinimide, 1,4-benzoquinone, and diethyl ketomalonate. While *N*-chlorosuccinimide and diethyl ketomalonate provide the desired products, enantioselectivity was very low with *N*-chlorosuccinimide. For this reason, diethyl ketomalonate was chosen as the optimal electrophile.

In addition, we further screened a range of electrophiles that had been employed in chiral phosphoric acid-catalyzed reactions. In these reactions, the desired products were not formed as shown below.

Entry	Electrophile	Time (h)	Yield ^a (%)	e.e. ^b (%)
1	2a	72	99	96
2	S39	72	32	3
3	S40	72	<5	-
4	S41	72	<5	-
5	S42	72	<5	-
6	S43	72	<5	-
7	S44	72	<5	-
8	S45	72	<5	-

^aIsolated yields. ^bEnantiomeric excesses were determined by chiral HPLC analysis.

In asymmetric catalysis, high selectivity can be achieved through the harmony between the substrates and the catalyst. In our computational study, such harmony is clearly observed that is π - π interactions between one of ester of electrophile with the indole ring and appropriate size of electrophile. Due to the absence of these factors, it is believed that other electrophiles were not compatible with our methodology. The results have been summarized in the Supplementary Information (Section 5.1.4), and the brief description has been added to the footnote as reference 62. Now, it reads.

“62. Electrophiles were screened and diethyl ketomalonate was chosen as the optimal electrophile. See the Supplementary Information for details.”

Overall, we agree the point raised by Reviewer 4 that demonstrating broad applicability is essential for our work to be considered for publication in *Nat. Comm.* This can certainly be achieved by showcasing compatibility with several electrophiles. However, we firmly believe that the broad applicability of our methodology is already evident from the high

enantioselectivity achieved in three different positions of stereogenic axes in indoles. To the best of our knowledge, no other methodology has been reported yet that is compatible with stereogenic axes at three different positions. For these compelling reasons, we are confident that our novel and groundbreaking research is well-suited for publication in *Nature Communications*."

Reviewer 4 Point 2: *For the methodology study: (b) For the asymmetric ketomalonates used in the work, the conformation of the newly formed chiral center should be demonstrated (for example compounds 3ad, 3id and 7dd). Also in this regard, more asymmetric ketomalonates should be studied.*

Our response: To determine the absolute structure, we attempted to get single crystal of both **3ld** and **7dd**. Ultimately, the single crystal X-ray structure of **3ld** was obtained, which allowed us to determine its absolute configuration (CCDC Deposition Number: 2284844).

Because the compounds **3** and **7** are in a reversed form of each other, they exhibit similar transition state structures as shown in Fig. 5d. In this regard, we believe that they share the same pattern of absolute configuration. The supposed absolute configuration for **7dd** has been designated in Fig. 4.

Furthermore, we successfully grew single crystals of racemic mixtures of **7dd**, which allowed us the X-ray crystal structure of (\pm)-**7dd**, as shown below. It also supports the relative configuration between the stereogenic axis and stereogenic center, depicted in Fig. 4.

In addition, we explored more asymmetric ketomalونات in our reaction.

N-1 Substrate: When the methyl or phenyl group was introduced instead of the trifluoromethyl group, the reaction did not proceed well. However, when the ethyl ester was replaced with the methyl ester, the desired product was obtained. Also, the reaction with ethyl 2-oxoacetate provides the substituted product albeit low selectivities. The new results are shown below.

^aThe reaction was performed at room temperature with 20 mol% of **P9**.

These results have been added to the main text in Page 11 and Fig. 3. Now, it reads.

“...The reaction of **1a** with methyl trifluoropyruvate (**2e**) yielded the desired product (**3ae**) in 6% yield, >20:1 dr, 92% ee under the optimal reaction conditions and 50% yield, >20:1 dr, 89% ee with 20 mol% of **P9** at room temperature. While methyl 2-oxoacetate (**2f**) produced the desired product albeit low selectivity (~1:1 dr, 73% ee), methyl 2-oxopropanoate (**2g**) and methyl 2-oxo-2-phenylacetate (**2h**) were found to be incompatible with our methodology. When the substrate with a 5-methoxy substitution (**1l**) was employed instead of **1a**, the reaction rate increased, but the selectivity was either

maintained or slightly lower. The reaction of 5-methoxy-*N*-arylidole (**1l**) with **2c** produced **3lc** with moderate enantioselectivity. The reaction of **1l** with **2d**, **2e**, and **2f** provided the desired product in higher yields and similar selectivities under the optimized reaction conditions. However, **2g** and **2h** still did not yield the desired product.”

C-2 and C-3 Substrate: While the substituted products were obtained from the reaction of **4a** with **2d** or **2e**, the selectivities were rather low. This result implies that the C-2 substrates would have distinct transition states from the N-1 and C-3 substrates. Also, we tested the unsymmetrical electrophile containing a methyl ester group (**2e**), which provided the desired product with excellent yield and moderate selectivity (99% yield, 9:1 dr, 77% ee).

In this process, we observed somewhat large difference in the enantioselectivities of **7dd** and **7de**. To investigate further, we re-performed the reaction of **6d** and **2d**, which resulted in slightly lower enantioselectivity (86% ee). Upon closer examination, we identified that an impurity in the racemic sample led to a misinterpretation in the analysis of the LC trace for **7dd**. We have since rectified this error in both the main manuscript and Supplementary Information.

These results have been added to the main text in Page 13, Page 15, and Fig. 4 left. Now, it reads.

“...The reaction of **4a** with unsymmetrical ketones (**1d** and **1e**) resulted in the desired product with excellent yield, but low selectivity.”

“...The reaction of 5-methoxy-3-arylidole (**6d**) with diisopropyl ketomalonate (**2c**) resulted in the desired product with 80% yield and 79% ee. The unsymmetrical electrophiles (**2d** and **2e**) were employed for the reaction of **6d**, which led to excellent yields and good to moderate enantioselectivities (>20:1 dr, 86% ee for **7dd** and 9:1 dr, 77% ee for **7de**).”

We appreciate the reviewer’s comment, which has provided valuable insights to expand the scope of our methodology. As a result, our work now showcases 46 examples demonstrating the control of stereogenic axes at three different positions of indoles.

Reviewer 4 Point 3: *For the methodology study (c) Have the authors tried other N substituent for the substrate 6 (Figure 4)? N-Benzyl and NH showed very poor results. How about N-ethyl and the N-propyl groups?*

Our response: According to the comment, *N*-ethyl and *N*-propyl substituted substrates were prepared and subjected to the optimal reaction conditions. It was observed that as the size of the substituent increased, the reaction rate became slower. With the optimal reaction conditions, the desired product was obtained with *N*-ethyl substrate in 5% yield and 40% ee. However, only a trace amount of product was formed with *N*-propyl substrate. To accelerate the reaction, the reactions were performed at 60 °C, which gave the *N*-ethyl product in 30% yield and 40% ee and the *N*-propyl product in 18% yield and 22% ee.

^aThe reaction was performed at 60 °C with toluene as a solvent.

These results have been added to the main text in Page 15 and Fig. 4 right (**7fa** and **7ga**). Now, it reads.

“...While the substrates in which the methyl group at the *N*-1 position was substituted with an ethyl group (**6f**) or propyl group (**6g**) produced the desired products, the enantioselectivities were significantly reduced (40% ee for **7fa** and 22% ee for **7ga**). Substitution with a benzyl group (**6h**) or the removal of the substitution (**6l**)...”

We believe that this is similar case to Reviewer 2 Point 2 and Reviewer 2 Point 4. Since the aromatic ring is substituted at the C-3 position and ketomalonate approaches the C-2 position, the *N*-1 position has no choice but to face the bulky substituents of the catalyst in the transition state. Due to this steric hindrance, we hypothesize that the reaction rate decreased as the size of the substituent at the *N*-1 position increases.

Reviewer 4 Point 4: *For the methodology study (d) There seems to be very limited space for medicinal chemistry modifications of the products generated by this method. A number of examples should be added to showcase such modifications.*

Our response: First and foremost, we sincerely appreciate the reviewer’s valuable comment concerning the medicinal chemistry modifications of our products. We carefully considered the comment and would like to clarify the primary focus of our work.

We would like to note that our work aims to provide new methodology that can control the stereogenic axes at the different positions of indoles. This unprecedented versatility allows us to enantioselectively synthesize three different types of indoles. We believe that our work would be a new pioneer in catalytic and atroposelective synthesis, because this concept can be extensively utilized different types of atropisomeric scaffolds.

While we acknowledge that many methodology papers demonstrate the derivatization of their products, which can be valuable in material science and drug discovery, it is essential to note that our work primarily aims to showcase the feasibility and practicality of our scaffold for future medicinal chemistry applications. In this context, we emphasize the demonstration of a certain level of biological activity, which highlights the potential of our methodology for future medicinal chemistry studies.

However, considering the concern of Reviewer 4, we have added derivatization of our product. As shown below, benzylation and oxidation were successfully conducted without any loss of enantioselectivity. We are confident that the other S_N2 -type alkylation would also be feasible, and the oxidized product can serve as an intermediate for further derivatization. Also, in the progress of the derivatization, 1 mmol scale reaction has been performed, which yielded the desired product in 74% yield and 96% ee.

These results have been added to the main text in Page 19 and Fig. 6c. Now, it reads.

“To further demonstrate the applicability of our methodology, a 1-mmol-scale reaction and derivatizations were conducted (Fig. 6c). The reaction of 1 mmol of **1a** with **2a** in the presence of **P9** resulted in the formation of the desired product in 74% yield and 96% ee. Subsequently, the product **3aa** underwent benzylation to yield compound **8** without any loss of enantioselectivity. In addition, the oxidation of **3aa** provided indole 3-carboxaldehyde compound (**9**), which can be further transformed into diverse compounds.”

Reviewer 4 Point 5: *For the antiproliferative activity study: (a) The authors used only four concentration points to produce the curves. This would lead to inaccuracy. At least eight points should be tested.*

Our response: According to the reviewer's comment, we conducted new experiments with eight data points and recalculated IC_{50} values with improved accuracy. The results have been added to Fig. 6b. Now, it shows.

<Fig. 6b>

Reviewer 4 Point 6: *For the antiproliferative activity study: (b) The compounds showed only moderate activity with IC_{50} values in the range of 10–20 μM . Such effects should not be claimed as “great efficacy” in the abstract. Regarding the antiproliferative activity, have the authors tested the compounds in normal mammalian cells to measure toxicity? Specificity needs to be demonstrated for the anticancer potential of the compounds.*

Our response: In this work, we have developed new methodology, which can control the stereogenic axes at three different positions of indoles. To show the applicability of the reaction products, the preliminary antiproliferative activities were screened, that showed 10–20 μM of IC_{50} against various cancer cells, even though extensive medicinal chemistry efforts had not been made. As the preliminary results, we thought that this result is meaningful, and provides an opportunity for further medicinal chemistry study to be conducted. However, in order to address any potential misunderstandings raised by Reviewer 4 Point 6, we have modified the term “great efficacy” to “good efficacy”. Now, it reads.

“...our new scaffold exhibits good efficacy in this regard.”

According to the second comment, the potent atropisomeric compounds (**3ma** and **ent-3ma**) were exposed to normal human lung fibroblast cell line (MRC-5) and antiproliferative activities were measured by the Sulforhodamine-B (SRB) method. Both

compounds showed 2~3 folds lower antiproliferative activities against normal cell line than cancer cell lines. (33.36 μM and 30.57 μM of IC_{50} respectively). These results have been added to the main text and Fig. 6. Now, it reads.

“...When both atropisomers were treated with normal human lung fibroblast cell line to test toxicity, they exhibited approximately 2 to 3 folds lower antiproliferative activities compared to cancer cells.”

We believe that further chemical biology study to find the target biomolecule and medicinal chemistry study would provide higher activity and lower toxicity compounds, and the compounds (**3ma** and *ent-3ma*) would be a good starting point for future study.

Reviewer 4 Point 7: For the antiproliferative activity study: (3) Compound stability should also be examined. For example, will the ester groups be hydrolyzed under the incubation conditions?

Our response: According to the reviewer’s comment, the compound **3aa** was exposed to DMEM medium for 72 h at 37 °C. As illustrated below, we could not observe any decomposition and the amount of **3aa** remained unchanged.

HPLC conditions: YMC-Pack SIL column (250 × 4.5 mm, S-5 μm , 12 nm), 5% *i*PrOH/Hx, 1 mL/min

For this result, the description has been added to the footnote as reference 75. Now, it reads.

“75. The compound **3aa** was exposed to DMEM medium for 72 h at 37 °C, in which any decomposition including hydrolysis was not observed.”

Reviewers' Comments:

Reviewer #2:

Remarks to the Author:

I am happy that all the problems are addressed and recommend it for publication in Nature Communications.

Reviewer #3:

Remarks to the Author:

The authors properly addressed all of my comments and suggestions. In particular, the quality of the computational part of this work has greatly improved. Overall, I think that the manuscript is now suitable for publication in this journal.

Reviewer #4:

Remarks to the Author:

The authors have conducted further studies to address the issues of the previous manuscript. Most of the issues have been successfully answered. For the new antiproliferative data, I would argue that the 2-3 fold difference between normal and tumor cells seems marginal. The authors should mention this limitation in the manuscript. With that corrected, and on the basis that the questions from all other referees have been successfully answered, I would recommend the publication of the revised manuscript.

With Respect to Reviewer 2:

Reviewer 2: *I am happy that all the problems are addressed and recommend it for publication in Nature Communications.*

Our response: We greatly appreciate the positive assessment.

With Respect to Reviewer 3:

Reviewer 3: *The authors properly addressed all of my comments and suggestions. In particular, the quality of the computational part of this work has greatly improved. Overall, I think that the manuscript is now suitable for publication in this journal.*

Our response: Once again, we would like to express our gratitude to Reviewer 3 for providing valuable advice that allowed us to achieve improved calculation results.

With Respect to Reviewer 4:

Reviewer 4: *The authors have conducted further studies to address the issues of the previous manuscript. Most of the issues have been successfully answered. For the new antiproliferative data, I would argue that the 2-3 fold difference between normal and tumor cells seems marginal. The authors should mention this limitation in the manuscript. With that corrected, and on the basis that the questions from all other referees have been successfully answered, I would recommend the publication of the revised manuscript.*

Our response: According to the Reviewer 4's comment, we have added the phrase in Page 19 of the Result section. Now, it reads,

“...Although the variation in antiproliferative activity between cancer and normal cell lines was not dramatically significant, this study highlighted the significance of atroposelective synthesis of a biologically relevant scaffold and dependable atroposelective strategy.”